# First is Better Than Last for Language Data Influence

**Chih-Kuan Yeh** *
Google Inc.
chihkuanyeh@google.com

**Ankur Taly**
Google Inc.
ataly@google.com

**Mukund Sundararajan**
Google Inc.
mukunds@google.com

**Frederick Liu**
Google Inc.
frederickliu@google.com

**Pradeep Ravikumar**
Carnegie Mellon University
Department of Machine Learning
pradeepr@cs.cmu.edu

## Abstract

The ability to identify influential training examples enables us to debug training data and explain model behavior. Existing techniques to do so are based on the flow of training data influence through the model parameters (Koh & Liang, 2017; Yeh et al., 2018; Pruthi et al., 2020). For large models in NLP applications, it is often computationally infeasible to study this flow through *all* model parameters, therefore techniques usually pick the last layer of weights. However, we observe that since the activation connected to the last layer of weights contains "shared logic", the data influenced calculated via the last layer weights prone to a "cancellation effect", where the data influence of different examples have large magnitude that contradicts each other. The cancellation effect lowers the discriminative power of the influence score, and deleting influential examples according to this measure often does not change the model's behavior by much. To mitigate this, we propose a technique called TracIn-WE that modifies a method called TracIn (Pruthi et al., 2020) to operate on the word embedding layer instead of the last layer, where the cancellation effect is less severe. One potential concern is that influence based on the word embedding layer may not encode sufficient high level information. However, we find that gradients (unlike embeddings) do not suffer from this, possibly because they chain through higher layers. We show that TracIn-WE significantly outperforms other data influence methods applied on the last layer significantly on the case deletion evaluation on three language classification tasks for different models. In addition, TracIn-WE can produce scores not just at the level of the overall training input, but also at the level of words within the training input, a further aid in debugging.

## 1 Introduction

Training data influence methods study the influence of training examples on a model's weights (learned during the training process), and in turn on the predictions of other test examples. They enable us to debug predictions by attributing them to the training examples that most influence them, debug training data by identifying mislabeled examples, and fixing mispredictions via training data curation. While the idea of training data influence originally stems from the study of linear regression (Cook & Weisberg, 1982), it has recently been developed for complex machine learning models like deep networks.

Prominent methods for quantifying training data influence for deep networks include influence functions (Koh & Liang, 2017), representer point selection (Yeh et al., 2018), and TracIn (Pruthi et al., 2020). While the details differ, all methods involves computing the gradients (w.r.t. the loss) of the model parameters at the training and test examples. Thus, they all face a common computational

---

* part of work done in CMU

36th Conference on Neural Information Processing Systems (NeurIPS 2022).

challenge of dealing with the large number of parameters in modern deep networks. In practice, this challenge is circumvented by restricting the study of influence to only the parameters in the last layer of the network. While this choice may not be explicitly stated, it is often implicit in the implementations of larger neural networks. In this work, we revisit the choice of restricting influence computation to the last layer in the context of large-scale Natural Language Processing (NLP) models.

We first introduce the phenomenon of "cancellation effect" of training data influence, which happens when the sum of the influence magnitude among different training examples is much larger than the influence sum. This effect increases the influence magnitude of most training examples and reduces the discriminative power of data influence. We also observe that different weight parameters may have different level of cancellation effects, and the weight parameters of bias parameters and latter layers may have larger cancellation effects. To mitigate the "cancellation effect" and find a scalable algorithm, we propose to operate data influence on weight parameters with the least cancellation effect – the first layer of weight parameter, which is also known as the word embedding layer.

While word embedding representations might have the issue of not capturing any high-level input semantics, we surprisingly find that the gradients of the embedding weights do not suffer from this. Since the gradient chain through the higher layers, it thus takes the high-level information captured in those layers into account. As a result, the gradients of the embedding weights of a word depend on both the context and importance of the word in the input. We develop the idea of word embedding based influence in the context of TracIn due to its computational and resource efficiency over other methods. Our proposed method, TracIn-WE, can be expressed as the sum of word embedding gradient similarity over overlapping words between the training and test examples. Requiring overlapping words between the training and test sentences helps capture low-level similarity, while the word gradient similarity helps capture the high-level semantic similarity between the sentences. A key benefit of TracIn-WE is that it affords a natural word-level decomposition, which is not readily offered by existing methods. This helps us understand which words in the training example drive its influence on the test example.

We evaluate TracIn-WE on several NLP classification tasks, including toxicity, AGnews, and MNLI language inference with transformer models fine-tuned on the task. We show that TracIn-WE outperforms existing influence methods on the case deletion evaluation metric by $4 - 10\times$. A potential criticism of TracIn-WE is its reliance on word overlap between the training and test examples, which would prevent it from estimating influence between examples that relate semantically but not syntactically. To address this, we show that the presence of common tokens in the input, such as a "start" and "end" token (which are commonly found in modern NLP models), allows TracIn-WE to capture influence between semantically related examples without any overlapping words, and outperform last layer based influence methods on a restricted set of training examples that barely overlaps with the test example.[2]

## 2 Preliminaries

Consider the standard supervised learning setting, with inputs $x \in \mathcal{X}$, outputs $y \in \mathcal{Y}$, and training data $\mathbf{D} = \{(x_1, y_1), (x_2, y_2), ...(x_n, y_n)\}$. Suppose we train a predictor $\mathbf{f}$ with parameter $\Theta$ by minimizing some given loss function $\ell$ over the training data, so that $\Theta = \arg\min_\Theta \sum_{i=1}^n \ell(\mathbf{f}(x_i), y_i)$. In the context of the trained model $\mathbf{f}$, and the training data $\mathbf{D}$, we are interested in the data importance of a training point $x$ to the testing point $x'$, which we generally denote as $\mathbf{I}(x, x')$.

### 2.1 Existing Methods

We first briefly introduce the commonly used training data influence methods: Influence functions (Koh & Liang, 2017), Representer Point selection (Yeh et al., 2018), and TracIn (Pruthi et al., 2020). We demonstrate that each method can be decomposed into a similarity term $S(x, x')$, which measures the similarity between a training point $x$ and the test point $x'$, and loss saliency terms $L(x)$ and $L(x')$, that measures the saliency of the model outputs to the model loss. The decomposition largely derives from an application of chain rule to the parameter gradients.

$$\mathbf{I}(x, x') = L(x)S(x, x')L(x')$$

The decomposition yields the following interpretation. A training data $x$ has a larger influence on a test point $x'$ if (a) the training point model outputs have high loss saliency, (b) the training point $x$ and the test point $x'$ are similar as construed by the model. In Section 3.3, we show that restricting

---

the influence method to operate on the weights in the last layer of the model critically affects the similarity term, and in turn the quality of influence. We now introduce the form of each method, and the corresponding similarity and loss saliency terms.

**Influence Functions:**

$$\text{Inf}(x, x') = -\nabla_\Theta \ell(x, \Theta)^T H_\Theta^{-1} \nabla_\Theta \ell(x', \Theta),$$

where $H_\Theta$ is the hessian $\sum_{i=1}^n \nabla_\Theta^2 \ell(x, \Theta)$ computed over the training examples. By an application of the chain rule, we can see that $\text{Inf}(x, x') = L(x)S(x, x')L(x')$, with the similarity term $S(x, x') = \frac{\partial \mathbf{f}(x, \Theta)}{\partial \Theta}^T H_\Theta^{-1} \frac{\partial \mathbf{f}(x', \Theta)}{\partial \Theta}$, and the loss saliency terms $L(x) = \frac{\partial \ell(x, \Theta)}{\partial \mathbf{f}(x, \Theta)}$. The work by Sui et al. (2021) is very similar to extending the influence function to the last layer to satisfy the representer theorem.

**Representer Points:**

$$\text{Rep}(x, x') = -\frac{1}{2\lambda n} \frac{\partial \ell(x, \Theta)}{\partial \mathbf{f}_j(x, \Theta)} a(x, \Theta)^T a(x', \Theta), \tag{1}$$

where $a(x, \Theta)$ is the final activation layer for the data point $x$, $\lambda$ is the strength of the $\ell_2$ regularizer used to optimize $\Theta$, and $j$ is the targeted class to explain. The similarity term is $S(x, x') = a(x, \Theta)^T a(x', \Theta)$, and the loss saliency terms are $L(x) = \frac{1}{2\lambda n} \frac{\partial \ell(x, \Theta)}{\partial \mathbf{f}_j(x, \Theta)}$, $L(x') = 1$.

**TracIn:**

$$\text{TracIn}(x, x') = -\sum_{c=1}^d \eta_c \nabla_{\Theta_c} \ell(x, \Theta_c)^T \nabla_{\Theta_c} \ell(x', \Theta_c), \tag{2}$$

where $\Theta_c$ is the weight at checkpoint $c$, and $\eta_c$ is the learning rate at checkpoint $c$. In the remainder of the work, in our notation, we suppress the sum over checkpoints of TracIn for notational simplicity. (This is not to undermine the importance of summing over past checkpoints, which is a crucial component in the working on TracIn.) For TracIn, the similarity term is $S(x, x') = \nabla_\Theta \mathbf{f}(x, \Theta)^T \nabla_\Theta \mathbf{f}(x', \Theta)$, while the loss terms are $L(x) = \frac{\partial \ell(x, \Theta)}{\partial \mathbf{f}(x, \Theta)}$, $L(x') = \frac{\partial \ell(x', \Theta)}{\partial \mathbf{f}(x', \Theta)}$.

## 2.2 Evaluation: Case Deletion

We now discuss our primary evaluation metric, called *case deletion diagnostics* (Cook & Weisberg, 1982), which involves retraining the model after removing influential training examples and measuring the impact on the model. This evaluation metric helps validate the efficacy of any data influence method in detecting training examples to remove or modify for targeted fixing of misclassifications, which is the primary application we consider in this work. This evaluation metric was also noted as a key motivation for influence functions (Koh & Liang, 2017). Given a test example $x'$, when we remove training examples with positive influence on $x'$ (*proponents*), we expect the prediction value for the ground-truth class of $x'$ to decrease. On the other hand, when we remove training examples with negative influence on $x'$ (*opponents*), we expect the prediction value for the ground-truth class of $x'$ to increase.

An alternative evaluation metric is based on detecting mislabeled examples via self-influence (i.e. influence of a training sample on that same sample as a test point). We prefer the case deletion evaluation metric, as it more directly corresponds to the concept of data influence. Similar evaluations that measure the change of predictions of the model after a group of points is removed is seen in previous works. Han et al. (2020) measures the test point prediction change after $10\%$ training data with the most and least influence are removed, and Koh et al. (2019) measures the correlation of the model loss change after a group of trained data is removed and the sum of influences of samples in the group, where the group can be seen as manually defined clusters of data.

**Deletion curve.** Given a test example $x'$ and influence measure $\mathbf{I}$, we define the metrics $\text{DEL}_+(x', k, \mathbf{I})$ and $\text{DEL}_-(x', k, \mathbf{I})$ as the impact on the prediction of $x'$ (for its groundtruth class) upon removing top-$k$ proponents and opponents of $x'$ respectively:

$$\text{DEL}_+(x', k, \mathbf{I}) = \mathbb{E}[f_c(x', \Theta_{+;k}) - f_c(x', \Theta)],$$
$$\text{DEL}_-(x', k, \mathbf{I}) = \mathbb{E}[f_c(x', \Theta_{-;k}) - f_c(x', \Theta)],$$

where, $\Theta_{+;k}$ ( $\Theta_{-;k}$) are the model weights learned when top-$k$ proponents (opponents) according to influence measure $I$ are removed from the training set, and $c$ is the groundtruth class of $x'$. The expectation is over the number of retraining runs. We expect $\text{DEL}_+$ to have large negative, and $\text{DEL}_-$ to have large positive values. To evaluate the deletion metric at different values of $k$, we may plot $\text{DEL}_+(x', k, \mathbf{I})$ and $\text{DEL}_-(x', k, \mathbf{I})$ for different values of $k$, and report the area under the curve (AUC): $\text{AUC-DEL}_+ = \sum_{k=k_1}^{k_m} \frac{1}{m} \text{DEL}_+(x', k, \mathbf{I})$, and $\text{AUC-DEL}_- = \sum_{k=k_1}^{k_m} \frac{1}{m} \text{DEL}_-(x', k, \mathbf{I})$.

We note that the *case deletion diagnostics* is different to the leave-one-out evaluation of Koh & Liang (2017) by two points. First, leave-one-out evaluation focuses on removing one point, which is more meaningful in the convex regime where the optimization is initialization-invariant. We consider the leave-k-out evaluation which is closer to actual applications, as one may need to alter more than one training data to fix a prediction. Second, we consider the expected value of leave-k-out, to hedge the variance caused by specific model states, which was pointed out by Søgaard et al. (2021) to be a major issue for leave-one-out evaluation (especially when the objective is no longer convex).

## 3 Cancellation Effect of Data Influence

The goal of a data influence method is to distribute the test data loss (prediction) across training examples, which can be seen as an attribution problem where each training example is an agent. We observe *cancellation* across the data influence attributions to training examples, i.e., the sign of attributions across training examples disagree and cancels each other out. This leads to most training examples having a large attribution magnitude, which reduces the discriminatory power of attribution-based explanations.

Our next observation is that the cancellation effect varies across different weight parameters. In particular, when a weight parameter is used by most of the training examples, the cancellation effect is especially severe. One such parameter is the bias, whose cancellation effect is illustrated by the following example:

**Example 3.1.** *Consider an example where the input $x \in \mathbb{R}^d$ is sparse, and $x_i$ has feature $i$ with value 1 and all other features with value 0. The prediction function has the form $\mathbf{f}(x) = x \cdot w + b$. It follows that a set of optimal parameters are $w_i = y_i, b = 0$. We further assume that the parameter $b$ is initialized to $0$ and has never changed during the gradient descent progress. In this case, it is clear that the bias parameter $b$ is irrelevant to the model (as removing it will not change the model at all). However, the influence to individual examples caused by the bias $b$ is still non-zero. This is because even that the sum of gradient for bias $b$ is 0, (s.t. $\sum_i \frac{\partial L(\mathbf{f}(x_i), y_i)}{\partial b} = 0$), each individual term $\frac{\partial L(\mathbf{f}(x_i), y_i)}{\partial b} = \frac{\partial L(\mathbf{f}(x_i), y_i)}{\partial \mathbf{f}(x_i)}$ is non-zero for most $x_i$. Note that $\frac{\partial L(\mathbf{f}(x_i), y_i)}{\partial b}$ contributes to the influence to data $x_i$ directly, and thus the bias parameter $b$ will contribute to the influence of all the training data. In the contrary, for each weight variable $w_j$, $\frac{\partial L(\mathbf{f}(x_i), y_i)}{\partial w_j}$ is only non-zero for $x_j$, and thus the weight variable $w_j$ only contributes to the influence of one training data $x_j$. Thus, the bias would affect the influence for more training examples compared to the weights.*

The above example illustrates that while the bias parameter is not important for the prediction model (removing the bias can still lead to the same optimal solution), the total gradient that flows through the bias still high. In fact, we find empirically that the total influence that flows through the bias is larger than that flowing through the weight, since each training example's gradient will affect the bias but the total contribution will be cancelled out, so the bias will remain 0. We also note that even for deep network models that do not have a sparse input, the neurons connected to the weight are often 0 (due to ReLU types of activation functions). Thus, the gradient of weight parameters is often sparser compared to the gradient of bias parameters, and thus bias parameters would often have stronger cancellations, which we validate empirically.

### 3.1 Measuring the Cancellation Effect

In the above example, we defined strong cancellation effect when some weight parameters does not change a lot during training (or has saturated in the training process), but the total strength of the gradient of the weight parameters summed over training data is large. For weights $W$, we first define two terms $\Delta W_c$ and $G(W)_c$,

$$\Delta W_c = \|W_{c+1} - W_c\|,$$

$$G(W)_c = \sum_{x_i, y_i \sim D} \eta_c \|\frac{\partial l(x_i, y_i)}{\partial W_c}\|,$$

where $\Delta W^c$ measures the norm of weight parameter change between checkpoint $c$ and $c + 1$, and $G(W)_c$ measures the sum of weight gradient norm times learning rate summed over all training data. When $\Delta W_c$ is small, this means that the weight $W$ may have saturated at checkpoint $c$, and the weight may not actually affect the model output much (and thus the weight $W$ is not important for this epoch of training). When $G(W)_c$ is large, this means that the sum of gradient norm with respect to $W_c$ is still large, and the influence norm caused by $\frac{\partial l(x_i, y_i)}{\partial W_c}$ will also be large.

To measure the cancellation effect, we define the cancellation ratio of a weight parameter $W$ as:

$$C(W) = \frac{\sum_c G(W)_c}{\sum_c \Delta W_c}.$$

When $G(W)_c$ is large and $\Delta W_c$ is small, this means that a non-important weight $W_c$ greatly influenced the total influence norm, which may be only possible if the influence contributed from $W_c$ to different examples cancelled each other out. Applying this interpretation to the cancellation of bias parameters, the intuition is that the bias parameters are not mainly responsible for the reduction of testing example loss change during the training process (since $\Delta W_c$ is small). However, they dominate the total influence strength due to their dense nature ($G(W)_c$ is large). Parameters with high cancellation may not be ideal to the calculation of influences.

## 3.2 Removing Bias In TracIn Calculation to Reduce Cancellation Effect

To investigate whether removing weights with high cancellation effect really helps improve influence quality, we conducted an experiment on a CNN text classification on Agnews dataset with $87\%$ test accuracy. The model is defined as follows: first a token embedding with dimension 128, followed by two convolution layers with kernel size 5 and filter size 10, one convolution layers with kernel size 1 and filter size 10, a global max pooling layer, and a fully connected layer; all weights are randomly initialized. The first layer is the token embedding, the second layer is the convolution layer, and the last layer is a fully connected layer. The model has 21222 parameters in total (excluding the token embedding), in which 102 parameters are bias variables. We find $C(\text{bias})$ to be 16789, and $C(\text{weight})$ to be 2555, which validates that the bias variables have a much stronger cancellation effect than the weight variables. A closer analysis shows that $G(\text{bias})$ is similar to $G(\text{weight})$ (627206 and 559142), but $\Delta(\text{bias})$ is much smaller than $\Delta(\text{weight})$ (0.74 and 4.37.) Even though the bias parameters has a much smaller total change compared to the weight parameters, their impact on the gradient norm (and thus influence norm) is even higher than the weight parameters. This verifies the intuition in Example 3.1 that the bias parameter has a stronger cancellation effect since the gradient to bias is almost activated for all examples despite the actual bias change being small. To further verify that the TracIn score contributed by the bias may lower the overall discriminatory power, we compute AUC-DEL$_+$ and AUC-DEL$_-$ for TracIn and TracIn-weight on AGnews with our CNN model. The AUC-DEL$_+$ for TracIn and TracIn-weight is $-0.036$ and $-0.065$ respectively, and the AUC-DEL$_+$ for TracIn and TracIn-weight is $0.011$ and $0.046$. The result shows that by removing the TracIn score contributed by the bias (with only 102 parameters), the overall influence quality improves significantly. Thus, in all future experiments, we remove the bias in calculation of data influence if not stated otherwise.

## 3.3 Influence of Latter Layers May Suffer from Cancellation

As mentioned in Section 1, for scalability reasons, most influence methods choose to operate only on the parameters of the last fully-connected layer $\Theta_{\text{last}}$. We argue that this is not a great choice, as the influence scores that stems from the last fully-connected weight layer may suffer from cancellation effect, as different examples "share logics" in the activation representation of this layer, and have a higher gradient similarity for different examples. Early layers, where examples have unique logic, may suffer less from the cancellation effect. We first measure the gradient similarity for different examples for each layer, which is $E_{x_a,x_b}\text{COS-SIM}[\partial l(x_a)/\partial w, \partial l(x_b)/\partial w]$, where COS-SIM is the cosine similarity. This measures the expected gradient cosine similarity between two examples. The expected gradient similarity for testing examples between different layers in the CNN classification are: first $0.035$, second $0.075$, third $0.21$, last $0.23$. This verifies that the latter layers in the neural network have more aligned gradients between examples, and thus share more logics between training examples. We report the cancellation ratio for each of the TracIn layer varaint in Table 1, where TracIn-first, TracIn-second, TracIn-third, TracIn-last, TracIn-All refer to TracIn scores based on weights of the first layer, second layer, third layer, last layer, and all layers (the bias is always omitted). As we suspected, early layers suffers less from cancellation, and latter layers suffers more from cancellation. To assess the impact on influence quality, we evaluate the AUC-DEL$_+$ and AUC-DEL$_-$ score for TracIn calculated with different layers on the AGnews CNN model in Tab. 1. We observe that removing examples based on influence scores calculated using parameters of later layers (with more "shared logic"") leads to worse deletion score compared to removing examples based on influence scores calculated using parameters of earlier layers (with more "unique logic"). Interestingly, the performance of TracIn-first even outperforms TracIn-all where all parameters are used. [3] We hypothesize that since the TracIn score based on later layers contain too much cancellation,

---

[3] We note that our investigation of last layer cancellation is limited to the setting when the whole model is trained to produce a single classification score, which may not hold in the setting where only the last layer is fine-tuned or tasks with a generative output.

Table 1: Cancellation Ratio and AUC-DEL table for various layers in CNN model in AGnews.

| Dataset | Metric | TR-first | TR-second | TR-third | TR-last | TR-all |
|---------|--------|----------|-----------|----------|---------|--------|
| AGnews | Cancellation $\downarrow$ | **1863** | 2019 | 3126 | 2966 | 2368 |
| | AUC-DEL+ $\downarrow$ | **−0.077** | −0.075 | 0.012 | −0.016 | −0.065 |
| | AUC-DEL− $\uparrow$ | **0.045** | 0.022 | 0.006 | −0.032 | **0.046** |

Table 2: Examples for word similarity for different examples containing word "not".

| Example | Premise | Hypothesis | Label |
|---------|---------|-----------|-------|
| S1 | I think he is very annoying. | I do **not** like him. | Entailment |
| S2 | I think reading is very boring. | I do **not** like to read. | Entailment |
| S3 | I think reading is very boring. | I do **not** hate burying myself in books. | Contradiction |
| S4 | She **not** only started playing the piano before she could speak, but her dad taught her to compose music at the same time. | She started to playing music and making music from very long ago. | Entailment |
| S5 | I think he is very annoying. | I don't like him. | Entailment |
| S6 | She thinks reading is pretty boring | She doesn't love to read | Entailment |
| S7 | She not only started playing the piano before she could speak, but her dad taught her to compose music at the same time | She started to playing music and making music from quite long ago | Entailment |

it is actually harmful to include these weight parameters in the TracIn calculation. In the following, we develop data influence methods by only using the first layer of the model, which suffers the least from cancellation effect.

# 4 Word Embedding Based Influence

In the previous section, we argue that using the latter layers to calculate influence may lead to the cancellation effect, which over-estimates influence. Another option is to calculate influence on all weight parameters, but may be computational infeasible when larger models with several millions of parameters are used. To remedy this, we propose operating on the first layer of the model, which contains the less cancellation effect since early layers encodes "unique logit". The first layer for language classification models is usually the word embedding layer in the case of NLP models. However, there are two questions in using the first layer to calculate data influence: 1. the word (token) embedding contains most of the weight parameters, and may be computational expensive 2. the word embedding layer may not capture influential examples through high-level information. In the rest of this section, we develop the idea of word embedding layer based training-data influence in the context of TracIn. We focus on TracIn due to challenges in applying the other methods to the word embedding layer: influence functions on the word embedding layer are computationally infeasible due to the large size (vocab size $\times$ embedding_dimension) of the embedding layer, and representer is designed to only use the final layer. We show that our proposed influence score is scalable thanks to the sparse nature of word embedding gradients, and contains both low-level and high-level information since the gradient to the word embedding layer can capture both high-level and low-level information about the input sentence.

## 4.1 TracIn on Word Embedding Layer

We now apply TracIn on the word embedding weights, obtaining the following expression:

$$\text{TracIn-WE}(x, x') = -\frac{\partial \ell(x, \Theta)}{\partial \Theta_{\text{WE}}}^T \frac{\partial \ell(x', \Theta)}{\partial \Theta_{\text{WE}}}, \tag{3}$$

Implementing the above form of TracIn-WE would be computationally infeasible as word embedding layers are typically very large (vocab size $\times$ embedding dimension). For instance, a BERT-base model has 23M parameters in the word embedding layer. To circumvent this, we leverage the sparsity of word embedding gradients $\frac{\partial \ell(x, \Theta)}{\partial \Theta_{\text{WE}}}$, which is a sparse vector where only embedding weights associated with words that occur in $x$ have non-zero value. Thus, the dot product between two word embedding gradients has non-zero values only for words $w$ that occur in both $x, x'$. With this observation, we can rewrite TracIn-WE as:

$$\text{TracIn-WE}(x, x') = -\sum_{w \in x \cap x'} \frac{\partial \ell(x)}{\partial \Theta_w}^T \cdot \frac{\partial \ell(x')}{\partial \Theta_w}, \tag{4}$$

where $\Theta_w$ are the weights of the word embedding for word $w$. We call the term $\frac{\partial \ell(x)}{\partial \Theta_w}^T \cdot \frac{\partial \ell(x')}{\partial \Theta_w}$ the *word gradient similarity* between sentences $x, x'$ over word $w$.

Table 3: Word Decomposition Examples for TracIn-WE

| | Sentence content | Label |
|---|---|---|
| Test Sentence 1 - T1 | I can always end my conversations so you would not get any answers because you are too lazy to remember anything | Toxic |
| Test Sentence 2 - T2 | For me, the lazy days of summer is not over yet, and I advise you to please kindly consider to end one's life, thank you | Toxic |
| Train Sentence - S1 | Oh yeah, if you're too lazy to fix tags yourself, you're supporting AI universal takeover in 2020. end it. kill it now. | Non-Toxic |
| | Word Importance | Total |
| TracIn-WE(S1, T1) | [S]: $-0.28$, [E]: $-0.07$, to: $-0.15$, **lazy**: $-7.6$, you: $-0.3$, end:$-0.3$, too:$-0.3$ | $-9.2$ |
| TracIn-WE(S1, T2) | [S]: $-0.17$, [E]: $-0.23$, to: $0.54$, lazy: $-0.25$, you: $0.25$, **end**: $-3.12$ | $-3.45$ |

### 4.2 Interpreting Word Gradient Similarity

Equation 4 gives the impression that TracIn-WE merely considers a bag-of-words style similarity between the two sentences, and does not take the semantics of the sentences into account. This is surprisingly not true! Notice that for overlapping words, TracIn-WE considers the similarity between gradients of word embeddings. Since gradients are back-propagated through all the intermediate layers in the model, they take into account the semantics encoded in the various layers. This is aligned with the use of word gradient norm $\|\frac{\partial \mathbf{f}(x)}{\partial \Theta_w}\|$ as a measure of importance of the word $w$ to the prediction $\mathbf{f}(x)$ (Wallace et al., 2019; Simonyan et al., 2013). Thus, word gradient similarity would be larger for words that are deemed important to the predictions of the training and test points.

Word gradient similarity is not solely driven by the importance of the word. Surprisingly, we find that word gradient similarity is also larger for overlapping words that appear in similar contexts in the training and test sentences. We illustrate this via an example. Table 2 shows 4 synthetic premise-hypothesis pairs for the Multi-Genre Natural Language Inference (MNLI) task (Williams et al., 2018). An existing pretrained model (He et al., 2020) predicts these examples correctly with softmax probability between $0.65$ and $0.93$. Notice that all examples contain the word 'not' once. The word gradient importance $\|\frac{\partial \mathbf{f}(x)}{\partial \Theta_w}\|$ for "not" is comparable in all 4 sentences. The value of word gradient similarity for 'not' is $0.34$ for the pair S1-S2, and $-0.12$ for S1-S3, while it is $-0.05$ for S1-S4. This large difference stems from the context in which 'not' appears. The absolute similarity value is larger for S1-S2 and S1-S3, since 'not' appears in a negation context in these examples. (The word gradient similarity of S1-S3 is negative since they have different labels.) However, in S4, 'not' appears in the phrase "not only ... but", which is not a negation (or can be considered as double negation). Consequently, word gradient similarity for 'not' is small between S1 and S4. In summary, we expect the absolute value of TracIn-WE score to be large for training and test sentences that have overlapping important words in similar (or strongly opposite) contexts. On the other hand, overlap of unimportant words like stop words would not affect the TracIn-WE score.

### 4.3 Word-Level Decomposition for TracIn-WE

An attractive property of TracIn-WE is that it decomposes into word-level contributions for both the testing point $x'$ and the training point $x$. As shown in (4), word $w$ in $x$ contributes to TracIn-WE$(x, x')$ by the amount $\frac{\partial \ell(x)}{\partial \Theta_w}^T \cdot \frac{\partial \ell(x')}{\partial \Theta_w} \mathbb{1}[w \in x']$; a similar word-level decomposition can be obtained for $x'$. Such a decomposition helps us identify which words in the training point $(x)$ drive its influence towards the test point $(x')$. For instance, consider the example in Table 3, which contains two test sentences (T1, T2) and a training sentence S1. We decompose the score TracIn-WE(S1, T1) and TracIn-WE(S1,T2) into words contributions, and we see that the word "lazy" dominates TracIn-WE(S1, T1), and the word "end" dominates TracIn-WE(S1, T2). This example shows that different key words in a training sentence may drive influence towards different test points. The feature-decomposition for influence introduces additional interpretability to why two examples are highly influenced. This is demonstrated in a case study where we cluster difficult training examples based on a normalized TracIn-WE score in Sec. A.

### 4.4 An approximation for TracIn-WE

As we note in Sec. 4.1, the space complexity of saving training and test point gradients scales with the number of words in the sentence. This may be intractable for tasks with very long sentences. We alleviate this by leveraging the fact that the word embedding gradient for a word $w$ is the sum of input word gradients from each position where $w$ is present. Given this decomposition, we can approximate the word embedding gradients by saving only the top-k largest input word gradients for

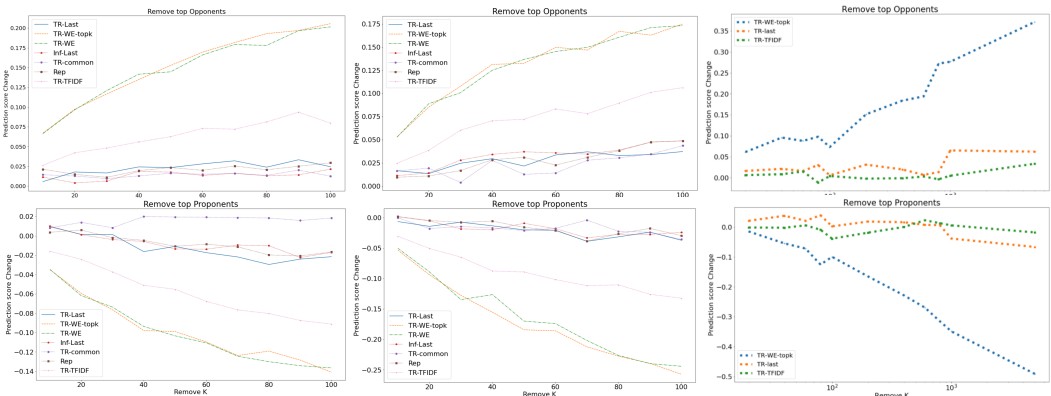

Figure 1: Deletion Curve for removing opponents (top figure, larger better) and proponents (bottom figure, smaller better) on Toxicity (left), AGnews (mid), and MNLI (right).

each sentence. (An alternative is to save the input word gradients that are above a certain threshold.) Formally, we define the approximation

$$\frac{\partial \ell(x, \Theta)}{\partial \Theta_w}\big|_{\text{top-k}} = \sum_{i \in x^{\text{top-k}} \wedge x^i = w} \frac{\partial \ell(x, \Theta)}{\partial x^i} \qquad (5)$$

where $x^i$ is the word at position $i$, and $x^{top-k}$ is the set of top-k input positions by gradient norm. We then propose

$$\text{TracIn-WE-Topk}(x, x) = - \sum_{w \in x \cap x'} \frac{\partial \ell(x, \Theta_w)}{\partial \Theta_w}\big|_{\text{top-k}}^T \cdot \frac{\partial \ell(x', \Theta_w)}{\partial \Theta_w}\big|_{\text{top-k}}. \qquad (6)$$

**Computational complexity** Let $L$ be the max length of each sentence, $d$ be the word embedding dimension, and $o$ be the average overlap between two sentences. If the training and test point gradients are precomputed and saved then the average computation complexity for calculating TracIn-WE for $m$ training points and $n$ testing points is $O(mnod)$. This can be contrasted with the average computation complexity for influence functions on the word embedding layer, which takes $O(mnd^2v^2 + d^3v^3)$, where $v$ is the vocabulary size which is typically larger than $10^4$, and $o$ is typically less than 5. The approximation for TracIn-We-Topk drops the computational complexity from $O(mnod)$ to $O(mno_kd)$ where $o_k$ is the average overlap between the sets of top-k words from the two sentences. It has the additional benefit of preventing unimportant words (ones with small gradient) from dominating the word similarity by multiple occurrences, as such words may get pruned. In all our experiments, we set $k$ to 10 for consistency, and do not tune this hyper-parameter.

### 4.5 Influence without Word-Overlap

One potential criticism of TracIn-WE is that it may not capture any influence when there are no overlapping words between $x$ and $x'$. To address this, we note that modern NLP models often include a "start" and "end" token in all inputs. We posit that gradients of the embedding weights of these tokens take into account the semantics of the input (as represented in the higher layers), and enable TracIn-WE to capture influence between examples that are semantically related but do not have any overlapping words. We illustrate this in S5-S7 in Tab. 2 via examples for the MNLI task. Sentence S5 has no overlapping words with S6 and S7. However, the word gradient similarity of "start" and "end" tokens for the pair S5-S6 is 1.15, while that for the pair S5-S7 is much lower at $-0.05$. Indeed, sentence S5 is more similar to S6 than S7 due to the presence of similar word pairs (e.g., think and thinks, annoying and boring), and the same negation usage. We further validate that TracIn-WE can capture influence from examples without word overlap via a controlled experiment in Sec. 5.

## 5 Experiments

We evaluate the proposed influence methods on 3 different NLP classification datasets with BERT models. We choose a transformer-based model as it has shown great success on a series of downstream tasks in NLP, and we choose BERT model as it is one of the most commonly used transformer model. For the smaller Toxicity and AGnews dataset, we operate on the Bert-Small model, as it already achieves good performance. For the larger MNLI dataset, we choose the Bert-Base model with $110M$ model parameters, which is a decently large model which we believe could represent the effectiveness of our proposed method on large-scale language models. As discussed in Section 2.2, we use the *case deletion* evaluation and report the metrics on the deletion curve in Table 4 for various methods and datasets. The standard deviation for all AUC values all methods is reported in Table 7.

Table 4: AUC-DEL table for various methods in different datasets. Highest number is bold.

| Dataset | Metric | Inf-Last | Rep | TR-last | TR-WE | TR-WE-topk | TR-TFIDF | TR-common |
|---|---|---|---|---|---|---|---|---|
| Toxic Bert | AUC-DEL+ ↓ | $-0.008$ | $-0.008$ | $-0.013$ | $\mathbf{-0.100}$ | $-0.099$ | $-0.067$ | $0.016$ |
| | AUC-DEL− ↑ | $0.014$ | $0.021$ | $0.023$ | $0.149$ | $\mathbf{0.151}$ | $0.063$ | $0.014$ |
| AGnews Bert | AUC-DEL+ ↓ | $-0.018$ | $-0.016$ | $-0.021$ | $-0.166$ | $\mathbf{-0.174}$ | $-0.090$ | $-0.017$ |
| | AUC-DEL− ↑ | $0.033$ | $0.028$ | $0.028$ | $0.130$ | $\mathbf{0.131}$ | $0.072$ | $0.023$ |
| MNLI Bert | AUC-DEL+ ↓ | | | $0.006$ | | $\mathbf{-0.198}$ | $-0.004$ | |
| | AUC-DEL− ↑ | | | $0.026$ | | $\mathbf{0.169}$ | $0.005$ | |
| Toxic Roberta | AUC-DEL+ ↓ | $-0.011$ | $-0.004$ | $0.001$ | $-0.030$ | $\mathbf{-0.038}$ | $-0.001$ | $-0.001$ |
| | AUC-DEL− ↑ | $0.023$ | $0.012$ | $0.003$ | $\mathbf{0.033}$ | $0.030$ | $0.006$ | $0.010$ |

| Dataset | Metric | Inf-Last | Rep | TR-last | TR-WE | TR-WE-topk | TR-WE-NoC | TR-common |
|---|---|---|---|---|---|---|---|---|
| Toxic Nooverlap | AUC-DEL+ ↓ | $-0.009$ | $-0.008$ | $-0.006$ | $\mathbf{-0.018}$ | $-0.016$ | $0.003$ | $-0.008$ |
| | AUC-DEL− ↑ | $0.008$ | $0.007$ | $0.010$ | $\mathbf{0.026}$ | $\mathbf{0.026}$ | $0.001$ | $0.015$ |

**Baselines** One question to ask is whether the good performance of TracIn-WE is a result that it captures the low-level word information well. To answer this question, we design a synthetic data influence score as the TF-IDF similarity Salton & Buckley (1988) multiplied by the loss gradient dot product for $x$ and $x'$. TR-TFIDF can be understood by replacing the embedding similarity of TracIn-Last by the TF-IDF similarity, which captures low level similarity.

$$\text{TR-TFIDF}(x, x') = -\text{Tf-Idf}(x, x')\frac{\partial \ell(x, \Theta)}{\partial \mathbf{f}(x, \Theta)}^T \frac{\partial \ell(x', \Theta)}{\partial \mathbf{f}(x, \Theta)}. \tag{7}$$

**Toxicity.** We first experiment on the toxicity comment classification dataset (Kaggle.com, 2018), which contains sentences that are labeled toxic or non-toxic. We randomly choose $50,000$ training samples and $20,000$ validation samples. We then fine-tune a BERT-small model on our training set, which leads to $96\%$ accuracy. Out of the $20,000$ validation samples, we randomly choose 20 toxic and 20 non-toxic samples, for a total of 40 samples as our targeted test set. For each example $x'$ in the test set, we remove top-$k$ proponents and top-$k$ opponents in the training set respectively, and retrain the model to obtain $\text{DEL}_+(x', k, \mathbf{I})$ and $\text{DEL}_-(x', k, \mathbf{I})$ for each influence method $\mathbf{I}$. We vary $k$ over $\{10, 20, \ldots, 100\}$. For each $k$, we retrain the model 10 times and take the average result, and then average over the 40 test points. We implement the methods Influence-last, Representer Points, TracIn-last, TracIn-WE, TracIn-WE-Topk, TracIn-TFIDF (introduced in Sec. G), TracIn-common (which is a variant of TracIn only using the start token and end token to calculate gradient), and abbreviate TracIn with TR in the experiments. We see that our proposed TracIn-WE method, along with its variants TracIn-WE-Topk outperform other methods by a significant margin. As mentioned in Sec. 3.3, TF-IDF based method beats the existing data influence methods using last layer weights by a decisive margin as well, but is still much worse compared to TracIn-WE. Therefore, TracIn-WE did not succeed by solely using low-level information. Also, we find that TracIn-WE performs much better than TracIn-common, which uses the start and end tokens only. This shows that the keyword overlaps (such as lazy, end in Table 3) is crucial to the great performance of TracIn-WE.

**AGnews.** We next experiment on the AG-news-subset (Gulli, 2015; Zhang et al., 2015), which contains a corpus of news with 4 different classes. We follow our setting in toxicity and choose $50,000$ training samples, $20,000$ validation samples, and fine-tune with the same BERT-small model that achieves $90\%$ accuracy on this dataset. We randomly choose 100 samples with 25 from each class as our targeted test set. The AUC-DEL$_+$ and AUC-DEL$_-$ scores for $k \in \{10, 20, \ldots, 100\}$ are reported in Table 4. Again, we see that the variants of TracIn-WE significantly outperform other existing methods applied on the last layer. In both AGnews and Toxicity, removing 10 top-proponents or top-opponents for TracIn-WE has more impact on the test point compared to removing 100 top-proponents or top-opponents for TracIn-last.

**MNLI.** Finally, we test on a larger scale dataset, Multi-Genre Natural Language Inference (MultiNLI) Williams et al. (2018), which consists of $433k$ sentence pairs with textual entailment information, including entailment, neutral, and contradiction. In this experiment, we use the full training and validation set, and BERT-base which achieves $84\%$ accuracy on matched-MNLI validation set. We choose 30 random samples with 10 from each class as our targeted test set. We only evaluate TracIn-WE-Topk, TracIn-last and TracIn-TFIDF as those were the most efficient methods to run at large scale. We vary $k \in \{20, \ldots, 5000\}$, and the AUC-DEL$_+$ and AUC-DEL$_-$ scores for our test set are reported in Table 4. Unlike previous datasets, here TracIn-TFIDF does not perform better

than TracIn-Last, which may be because input similarity for MNLI cannot be merely captured by overlapping words. For instance, a single negation would completely change the label of the sentence. However, we again see TracIn-WE-Topk significantly outperforms TracIn-Last and TracIn-TFIDF, demonstrating its efficacy over natural language understanding tasks as well. This again provides evidence that TracIn-WE can capture both low-level information and high-level information. The deletion curve of Toxicity, AGnews, MNLI is in shown in Fig. 4.5 and Fig. 1.

**Toxicity-Roberta.**   To additionally test whether our experiment results apply to more modern models, we repeat our experiments on the toxicity dataset with a Roberta model Liu et al. (2019), while fixing other settings. We find that the TracIn-WE and TracIn-WE-Topk still significantly outperforms other results.

**No Word Overlap.**   To assess whether TracIn-WE can do well in settings where the training and test examples do not have overlapping words, we construct a controlled experiment on the Toxicity dataset. We follow all experimental setting for Toxicity classification with the Bert model, but making two additional changes – (1) given a test sentence $x'$, we only consider the top-5000 training sentences (out of $50,000$) with the least word overlap for computing influence. We use TF-IDF similarity to rank the number of word overlaps so that stop word overlap will not be over-weighted. (2) We also fix the token embedding during training (result when word-embedding is not fixed is in the appendix, where removing examples based on any influence method does not change the prediction), as we find sentence with no word overlaps carry more influence when the token embedding is fixed. The AUC-DEL$_+$ and AUC-DEL$_-$ scores are reported in the lower section of Table 4. We find that TracIn-WE variants can outperform last-layer based influence methods even in this controlled setting, showing that TracIn-WE can retrieve influential examples even without non-trivial word overlaps. In Section 4.5, we claimed that this gain stems from the presence of common tokens ("start", "end", and other frequent words). To validate this, we compared with a controlled variant, TracIn-WE-NoCommon (TR-WE-NoC) where the common tokens are removed from TracIn-WE. As expected, this variant performed much worse on the AUC-DEL$_+$ and AUC-DEL$_-$ scores, thus confirming our claim. We also find that the result of TracIn-WE is better than TracIn-common (which is TracIn-WE with only "start" and "end" tokens), which shows that the common tokens such as stop words and punctuation may also help finding influential examples without meaningful word overlaps.

## 6   Related Work

In the field of explainable machine learning, our works belongs to training data importance (Koh & Liang, 2017; Yeh et al., 2018; Jia et al., 2019; Pruthi et al., 2020; Khanna et al., 2018; Sui et al., 2021). Other forms of explanations include feature importance feature-based explanations, gradient-based explanations (Baehrens et al., 2010; Simonyan et al., 2013; Zeiler & Fergus, 2014; Bach et al., 2015; Ancona et al., 2018; Sundararajan et al., 2017; Shrikumar et al., 2017; Ribeiro et al., 2016; Lundberg & Lee, 2017; Yeh et al., 2019; Petsiuk et al., 2018) and perturbation-based explanations (Ribeiro et al., 2016; Lundberg & Lee, 2017; Yeh et al., 2019; Petsiuk et al., 2018), self-explaining models (Wang & Rudin, 2015; Lee et al., 2019; Chen et al., 2019), counterfactuals to change the outcome of the model (Wachter et al., 2017; Dhurandhar et al., 2018; Hendricks et al., 2018; van der Waa et al., 2018; Goyal et al., 2019), concepts of the model (Kim et al., 2018; Zhou et al., 2018). For applications on applying data importance methods on NLP tasks, there have been works identifying data artifacts (Han et al., 2020; Pezeshkpour et al., 2021) and improving models (Han & Tsvetkov, 2020, 2021) based on existing data importance method using the influence function or TracIn. In this work, we discussed weight parameter selection to reduce cancellation effect for training data attribution. There has been works that discuss how to cope with cancellation in the context of feature attribution: Liu et al. (2020) discusses how regularization during training reduces cancellation of feature attribution, Kapishnikov et al. (2021) discusses how to optimize IG paths to minimize cancellation of IG attribution, and Sundararajan et al. (2019) discusses improved visualizations to adjust for cancellation.

## 7   Conclusion

In this work, we revisit the common practice of computing training data influence using only last layer parameters. We show that last layer representations in language classification models can suffer from the cancellation effect, which in turn leads to inferior results on influence. We instead recommend computing influence on the word embedding parameters, and apply this idea to propose a variant of TracIn called TracIn-WE. We show that TracIn-WE significantly outperforms last versions of existing influence methods on three different language classification tasks for several models, and also affords a word-level decomposition of influence that aids interpretability.

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
