## A    Limitations of Our Work

While we believe that our claim that "first is better than last for training data influence" is general, we did not test out the method on all data modalities and all types of models, as the computation of deletion score is very expensive. In this work, we focus on tasks of language classification, and mainly focus on BERT models (and an additional CNN model). We did not test on other types of language model, mainly because the amount of computation resource that is needed to test on different language model is already very expensive (see computation section). Furthermore, we expect models with similar architecture to have similar behaviors on data influence tasks, and the largest BERT-base model has similar parameters to similar architectures such as XLNET, RoBERTA. We note that we have not tested the last layer similarity for generative tasks, as it is beyond the scope of the paper, and we leave it to future works.

## B    Potential Social Impact of Our Work

One potential social impact is that one may use the algorithm to adjust training data to effect a particular test point's prediction. This can be used for good (making the model more fair), or for bad (making the model more biased).

## C    Computation

We report the run time for TracIn-WE, TracIn-Sec, TracIn-Last, Inf-Sec, Inf-Last for the CNN text model. The second convolutional layer has $6400$ parameters, last layer has $440$ parameters, token embedding layer has 3.2 million parameters. We applied these methods on $50000$ training points and $10$ test points. The preprocessing time (sec) per training point is $0.004, 0.004, 0.003, 3.52, 0.002$, and the cost of computing influence per training point and test point pair (sec) is: $4 \cdot 10^{-4}, 3 \cdot 10^{-5}$, $8 \cdot 10^{-6}, 10^{-1}, 2 \cdot 10^{-5}$. Influence function on the second layer is already order of magnitudes slower than other variants, and cannot scale to the word embedding layer with millions of parameters.

For remove and retrain on Toxicity and AGnews, we run our experiments on multiple V100 clusters. For remove and retrain on MNLI, we run our experiments on multiple TPU-v3 clusters. For toxicity and AGnews experiment, we need to fine-tune the language model on the classification task for $40 \times 6 \times 10 \times 2 \times 10$ times, where the fine-tuning takes around $10 - 20$ GPU-minute on a V100 for Bert-Small, and $40, 6, 10, 2, 10$ stands for number of test points, number of methods, removal numbers, proponents/ opponents, and repetition numbers respectively. On MNLI, we fine-tuned the language model for $19800(30 \times 3 \times 11 \times 2 \times 10)$ times, where fine-tuning MNLI on BERT-Base takes around 320 TPU-minute on a TPU-v3 cluster for Bert-base.

## D    Licence of Datatset

Toxicity dataset has license cc0-1.0, AGnews dataset has license non-commercial use, and MNLI has license cc-by-3.0.

## E    A different viewpoint on Issues with Last Layer.

We present our analysis in the context of the TracIn method applied to the last layer, referred to as TracIn-Last, although our experiments in Section 5 suggest that Influence-Last and Representer-Last may also suffer from similar shortcomings. For TracIn-Last, the similarity term $S(x, x') = \nabla_\Theta \mathbf{f}(x, \Theta_{last})^T \nabla_\Theta \mathbf{f}(x', \Theta_{last})$ becomes $a(x, \Theta_{last})^T a(x', \Theta_{last})$ where $a(x, \Theta_{last})$ is the final activation layer. We refer to it as *last layer similarity*. Overall, TracIn-last has the following formullation:

$$\text{TracIn-Last}(x, x') = a(x, \Theta)^T a(x', \Theta) \frac{\partial \ell(x', \Theta)}{\partial \mathbf{f}(x, \Theta)}^T \frac{\partial \ell(x', \Theta)}{\partial \mathbf{f}(x, \Theta)}.$$

We begin by qualitatively analyzing the influential examples from TracIn-Last, and find the top proponents to be unrelated to the test example. We also observe that the top proponents of different test examples coincide a lot; see appendix E for details. This leads us to suspect that the top influence scores from TracIn-Last are dominated by the loss salience term of the training point $x$ (which is

independent of $x'$), and not as much by the similarity term, which is also observed by Barshan et al. (2020); Hanawa et al. (2021). Indeed, we find that on the toxicity dataset, the top-100 examples ranked by TracIn-Last and the top-100 examples ranked by the loss salience term $\frac{\partial \ell(x,\Theta)}{\partial \mathbf{f}(x,\Theta)}$ have 49 overlaps on average, while the top-100 examples by TracIn-Last and the top-100 examples ranked by the similarity term $a(x,\Theta)^T a(x',\Theta)$ have only 22 overlaps on average. Finally, we find that replacing the last-layer similarity component by the well-known TF-IDF significantly improves its performance on the case deletion evaluation. In fact, this new method, which we call TracIn-TDIDF, also outperforms Influence-Last, and Representer-Last on the case deletion evaluation; see Section 5 and Appendix E. We end this section with the following hypothesis.

**Hypothesis E.1.** TracIn-Last and other influence methods that rely on last layer similarity fail in finding influential examples since last layer representations are too reductive and do not offer a meaningful notion of sentence similarity that is essential for influence.

We begin by qualitatively examining the influential examples obtained from TracIn-Last. Consider the test sentence and its top-2 proponents and opponents in Table 5. As expected, the proponents have the same label as the test sentence. However, besides this label agreement, it is not clear in what sense the proponents are similar to the test sentence. We also observe that out of 40 randomly chosen test examples, proponent-1 is either in the top-20 proponents or top-20 opponents for 39 test points.

Table 5: Examples for TracIn-Last

|  | Sentence content | Label |
|---|---|---|
| Test Sentence | Somebody that double clicks your nick should have enough info but don't let that cloud your judgement! There are other people you can hate for no reasons whatsoever. Hate another day. | Non-Toxic |
| Proponent-1 | Wow! You really are a piece of work, aren't you pal? Every time you are proven wrong, you delete the remarks. You act as though you have power, when you really don't. | Non-Toxic. |
| Proponent-2 | Ok i am NOT trying to piss you off ,but dont you find that touching another women is slightly disgusting. with all due respect, dogblue | Non-Toxic |
| Opponent-1 | Spot, grow up! The article is being improved with the new structure. Please stop your nonsense. | Toxic |
| Opponent-2 | are you really such a cunt? (I apologize in advance for certain individuals who are too sensitive) | Toxic |

To further validate that the inferior results from TracIn-Last can be attributed to the use of last layer similarity, we perform a controlled experiment where we replace the similarity term by a common sentence similarity measure — the TF-IDF similarity Salton & Buckley (1988).

$$\text{TR-TFIDF}(x, x') = -\text{Tf-Idf}(x, x') \frac{\partial \ell(x, \Theta)}{\partial \mathbf{f}(x, \Theta)}^T \frac{\partial \ell(x', \Theta)}{\partial \mathbf{f}(x, \Theta)}$$

We find that TFIDF performs much better than TracInCP-last and Influence-Last on the Del+ and Del- curve (see Fig. 1. This shows that last layer similarity does not provide a useful measure of sentence similarity for influence.

Since TF-IDF similarity captures sentence similarity in the form of low-level features (i.e., input words), we speculate that last layer representations are too reductive and do not preserve adequate low-level information about the input, which is useful for data influence. This is aligned with existing findings that last layer similarity in Bert models does not offer a meaningful notion of sentence similarity Li et al. (2020), even performing worse than GLoVe embedding.

# F  A Relaxation to Synonym Matching

While common tokens like "start" and "end" allow TracIn-WE to implicitly capture influence between sentences without word-overlap, the influence cannot be naturally decomposed over words in the two sentences. This hurts interpretability. To remedy this, we propose a relaxation of TracIn-WE, called TracIn-WE-Syn, which allows for synonyms in two sentences to directly affect the influence score. In what follows, we define synonyms to be words with similar embeddings.

Table 6: AUC-DEL table for various methods Toxicity with no overlap and embedding not fixed.

| Dataset | Metric | TR-last | TR-WE | TR-WE-topk | TR-WE-Syn | TR-WE-NoC |
|---------|--------|---------|-------|------------|-----------|-----------|
| Toxic | AUC-DEL+ $\downarrow$ | 0.001 | 0.002 | 0.003 | 0.004 | 0.006 |
| Nooverlap | AUC-DEL- $\uparrow$ | $-0.013$ | $-0.003$ | $-0.007$ | $-0.008$ | $-0.004$ |

We first rewrite word gradient similarity as

$$\text{WGS}_{x,x'}(w, w') = \frac{\partial \ell(x, \Theta)}{\partial \Theta_w}^T \frac{\partial \ell(x', \Theta)}{\partial \Theta_{w'}} \mathbb{1}[w = w'].$$

TracIn-WE can then be represented in the following form:

$$\text{TracIn-WE}(x, x') = -\sum_{w \in x} \sum_{w' \in x'} \text{WGS}_{x,x'}(w, w').$$

which can be seen as the sum of word gradient similarities for matching words in the two sentences. It is then natural to consider the variant where exact match is relaxed to synonym match:

$$\text{WGS-syn}_{x,x'}(w, w') = \frac{\partial \ell(x, \Theta)}{\partial \Theta_w}^T \frac{\partial \ell(x', \Theta)}{\partial \Theta_{w'}} \mathbb{1}[\text{Syn}(w, w') = 1].$$

where $\text{Syn}(w, w') = 1$ if the cosine similarity of the embeddings of $w$ and $w'$ is above a threshold. We set the threshold to be $0.7$ in our experiments. However, this direct relaxation has the caveat that a word $w$ in $x$ may be matched to several synonyms (including itself) in $x'$ simultaneously, which is not in the spirit of TracIn-WE where each word should only be matched to at most one word. To resolve this, we seek an optimal 1:1 match between words between the two sentence that respects synonymy and maximizes influence. We formulate this in terms of the Monge assignment problem (Peyré et al., 2019) from optimal transport. For scalability reasons, we operate on the top-$k$ relaxation of TracIn-WE (Section 4.4). Let $\{w_1, w_2, ...w_k\}$ and $\{w'_1, w'_2, ...w'_k\}$ be the top-$k$ words contained in $x$ and $x'$ respectively. Our goal is to find the optimal assignment function $m \in \mathbb{M} : \{1, ..., k\} \to \{1, ..., k\}$, such that $m(i) \neq m(j)$ for $i \neq j$ where

$$m^* = \arg \min_{m \in \mathbb{M}} \sum_{i=1}^{k} -|\text{WGS-syn}_{x,x'}(w_i, w'_{m(i)})|. \tag{8}$$

We define the matching cost between $w$ and $w'$ to be the negative absolute value of the word gradient similarity, as this allows us to match synonyms with strong positive as well as strong negative influence. Optimal assignment can be calculated efficiently by existing solvers, for instance, linear_sum_assignment function in SKlearn (Pedregosa et al., 2011). The final total influence can be obtained by

$$\text{TracIn-WE-Syn}(x, x) = -\sum_{w_i \in x} \text{WGS-syn}_{x,x'}(w_i, w'_{m^*(i)}),$$

We report the result for this relaxation in the following table 7, the result for TR-WE-Syn is close to the result of TracIn-WE, hinting that the additional synonym matching is not particular helpful for the deletion evaluation.

# G   Qualitative Examples

We show qualitative examples of the top-proponents and top-opponents for two random test points on dataset Toxicity (Tab. 8, 9), AGnews (Tab. 10, 11), and MNLI (Tab. 12, 13).

# H   No word overlap Experiment – More Details

**Why Fix Word Embedding:**   We first start by the conclusion of our observation: many influence methods cannot find training examples that influences a test point without word overlap in the case

Table 7: AUC-DEL table for various methods in different datasets.

| Dataset | Metric | Inf-Last | Rep | TR-last | TR-WE | TR-WE-topk | TR-WE-Syn | TR-TFIDF |
|---|---|---|---|---|---|---|---|---|
| Toxic | AUC-DEL+ ↓ | −0.022 | −0.021 | −0.025 | **−0.105** | −0.104 | −0.103 | −0.067 |
|  | AUC-DEL− ↑ | −0.001 | 0.006 | 0.007 | 0.122 | **0.125** | **0.125** | 0.044 |
| AGnews | AUC-DEL+ ↓ | −0.025 | −0.021 | −0.032 | −0.148 | **−0.152** | −0.142 | −0.083 |
|  | AUC-DEL− ↑ | 0.023 | 0.021 | 0.017 | **0.100** | **0.100** | 0.096 | 0.054 |

| Dataset | Metric | Inf-Last | Rep | TR-last | TR-WE | TR-WE-topk | TR-WE-Syn | TR-WE-NoC |
|---|---|---|---|---|---|---|---|---|
| Toxic | AUC-DEL+ ↓ | −0.011 | −0.015 | −0.007 | **−0.033** | −0.016 | −0.026 | 0.005 |
| Nooverlap | AUC-DEL− ↑ | 0.013 | 0.012 | 0.013 | **0.043** | 0.042 | 0.035 | −0.002 |

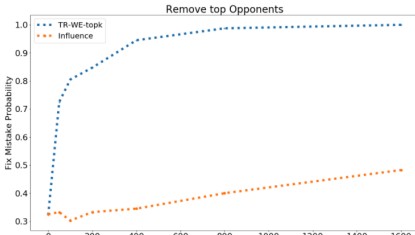
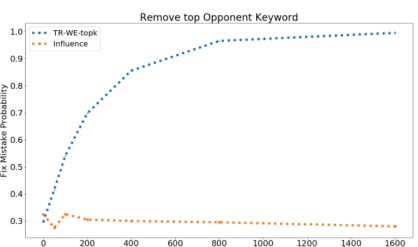

Figure 2: Probability to fix a mistake on Toxicity dataset by removing opponents and the removing one key word in opponents.

where word embedding is not fixed. To support this observation, we show the deletion curve on no word overlap experiment (when word embedding is not fixed during training) in Fig. 4 and the AUC-DEL score in Tab. 6. We can see after that removing proponents the Deletion score is actually slightly positive for all methods, and that removing opponents the Deletion score is actually slightly negative for all methods. This shows that no influence methods is able to find training examples that influence the test point without having word overlaps.

We thus suspect that influence may flow through examples pairs without word overlaps when the embedding is fixed. The intuition is that if you have two words A and A', that have the same initial word embedding. When embedding is not fixed, the embedding of A' and A may grow apart during training. However, if the embedding is fixed, the input of A and A' will always be the same regardless of whether if the training is applied on A and A'. Based on this intuition, we fix the word embedding during the model training for the no word overlap experiment. We now show the deletion curve for our experiment on no word overlap (when word embedding is fixed during training) in Fig. 3 (which is omitted from main text due to space constraint). We observe that although the signal is weak, most methods other than TR-WE-Noc is consistently positive when opponents are removed, and

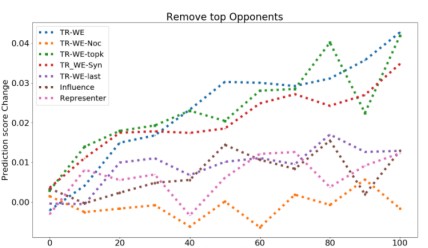
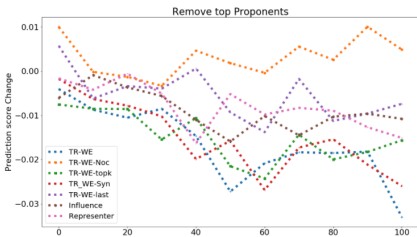

Figure 3: Deletion Curve or Toxicity dataset for removing opponents (larger better) and the removing proponents (smaller better).

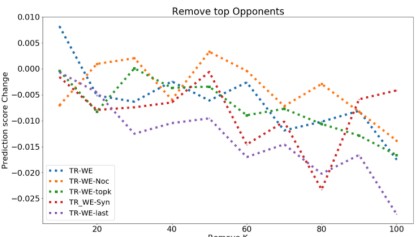
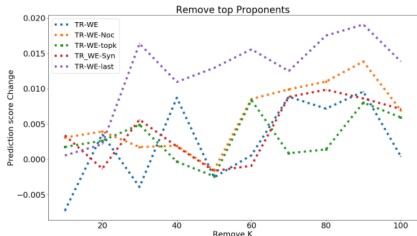

Figure 4: Deletion Curve or Toxicity dataset for removing opponents (larger better) and the removing proponents (smaller better).

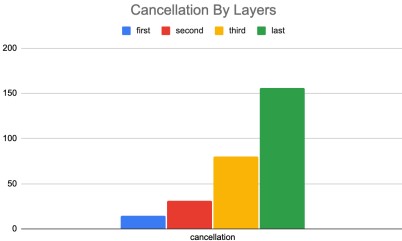

Figure 5: Cancellation Figure.

consistently negative when proponents are removed. As our result of AUC-DEL suggests, TR-WE variants perform the best in this case.

# I  Other Experiment Details

For Toxicity and AGnews, we use the small-Bert model[1] as our base model and fine-tune on our validation set. For MNLI, we use normal Bert models[2] and fine-tune on the validation set. For checkpoint selection, we follow suggestions in Pruthi et al. (2020) and choose $3 - 5$ checkpoints where the loss has not saturated yet. We follow standard fine-tuning procedures using SGD optimizers with momentum $0.9$, and we fine-tune for 10 epochs on AGnews with $2e^{-2}$ learning rate and fine-tune for 20 epochs on Toxicity with $2e^{-4}$ learning rate. The retraining parameters is fixed during the calculation of deletion curve. We split the training and validation set randomly and fix the random seed.

For MNLI, we calculate deletion curve for k $\in [20, 40, 60, 80, 100, 200, 400, 600, 800, 1000, 5000]$, and we can see from Fig. 1 that removing 60 examples based on TracIn-WE-Syn affects the test point more than removing 5000 examples based on TracIn-Last. The fine-tuning of MNLI follows standard framework in Tensorflow model garden (Yu et al., 2020).

We also clarify that in the context of our work, we refer to the tokens and words interchangeably for presentation simplicity. In our work, we use the tokenizer that is used along with Bert, which contains mostly words but also some word piece. When using a character-based tokenizer, the usage of "word" would then become characters.

# J  Plot of Cancellation

We add a plot for cancellations for different layers in a CNN model for Toxic classification in Fig. 5.

# K  Targeted Fixing of Misclassifications

We now discuss an application of our influence method in fixing specific misclassifications made by the model. We propose two means of fixing (a) remove top-k opponents (b) replace the most negatively influential word in each of the top-k opponents by [PAD]. The most influential word may

---

[1]https://huggingface.co/google/bert_uncased_L-2_H-128_A-2

[2]https://huggingface.co/bert-base-uncased

be identified using the word-level decomposition of TracIn-WE; see Section 4.3. We consider a
BERT model for the toxicity comment classification task Kaggle.com (2018), and randomly chose 40
misclassifications from the test set with prediction probability in $[0.3, 0.7]$. For each misclassification,
we apply the two approaches mentioned above for various values of $k$. For each $k$, we report the
average percentage of the mistake being fixed in 10 rounds of retraining.

We compare TracIn-WE-Topk with Influence-Last. To identify the most influential word using
Influence-Last, we consider its gradient w.r.t. to each word, which is suggested in a similar use case
by Pezeshkpour et al. (2021). For fix method (a), removing 50 opponents by TracIn-WE-Topk can fix
a mistake $73\%$ of the time, while removing 50 opponents by Influence-last can only fix it $33\%$ of
the time. With both methods, the average accuracy of the model after removing 50 examples only
drops by $0.01\%$. For fix method (b), removing the most negatively influential word for the top-200
opponents by TracIn-WE-Topk can fix a mistake $70\%$ of the time, while the same for Influence-Last
can only fix a mistake $30\%$ of the time. With both methods, the average accuracy of the model after
removing the most negatively influential word in the top-200 opponents drops by less than $0.02\%$.

We show the full fixing curve in the fixing application in Fig 2, where x-axis is the number of
training sentence we remove (either full remove or only remove one top key word). We show that
TrackIn-WE-topk significantly outperforms Influence-last in the targeted fixing application across
different number of removal $k$. When remove num $k = 0$, we see that the fix probability is 0.3,
meaning that after direct retrain without removal, the mistake can actually be correctly classified by
the model $30\%$ of the time.

# L    Exploration on Second Layer

One interesting follow-up is whether the second layer could be a better choice compared to the first
layer. While this is not in the main scope of our paper, we have tested this on the Toxicity dataset.
TracIn-Second is defined as using only the second layer parameters to calculate TracIn. Our results
show that TR-Second achieves AUC-DEL+ score of $-0.031$ and AUC-DEL$-$ score of $0.017$ in
Toxicity. This result is worse than TracIn-We and TR-TFIDF but better than TR-Last (see Tab. 4).
Therefore, this initial result shows that first is not only better than last, but is also better than second.

Table 8: Proponents and opponents for TracIn-Last on Toxicity

|  | Sentence content | Label |
|---|---|---|
| Test Sentence | I find Sandstein's dealing with the Mbz1 phenomenon very professional. He removed the soapbox image from that user's page and also banned you for not complying with your topic ban. It is you the one who is not assimilating the teaching of your topic ban. For example. You are topic banned because you don't have a professional approach to I-P topic and in general to any topic related to Jews and Judaism. The most resent example. When you reported that soapbox you qualified it as antisemitic. You at least should get informed of what that is. A neutral approach would be to have called it as soapbox canvasing and that's it. You should focus in your pictures which is the thing that you manage to do relatively well. Once you get into your holly war program of fighting all that in your imagination is an attack to Judaism you simply behave stupidly. It is those kinds of behaviors the ones that keep bringing hatred to us. That kind of attitude is, know it, racist, and if you are true to the struggles of the people of Abraham you above all should regret behaving as a racist. Once more, focus on your pictures and maybe even Sandstein will take a like on you. | Non-Toxic |
| Proponent-1 | You mean my past BLOCK. The third block was because of your incompetence. Jesus doesn't like liars. | Non-Toxic. |
| Proponent-2 | Pontiac Monrana Karrmann you full of shit ibelive all of the people who know that the montan will return after 2008 and we want ot knwo ehre do you get your info form and can you sohw it and guess what you dont know anythng about the Montana | Non-Toxic |
| Opponent-1 | I doubt this will get through your thick head (it's not an insult, it's an opinion based on your response) but the problem is not the issue itself. It's that people like you seem to enjoy (whether or not your side gets it right) to discuss, turn, twist and frankly abuse topics like this which are detrimental to the basic goals of Wikis in general and Wikipedia in particular. As John Stewart said to two hacks; You're hurting us. | Toxic |
| Opponent-2 | Pss Rex, be sure to DOCUMENT all the things you've discovered on the John Kerry page etc. It's awesome that you and I INDEPENDENTLY observed (and can corrorborate) virtually the exactsame pattern by these liberals. Demonizing conservatives; lionizing liberals. It's repeated ad infinitum, ad nauseum. The more proof we have, the easier it will be to persuade all but their fellow brain-dead truth haters to give a red cent to Wikipedia. And, until WHOLESALE changes are made from the top down, that's exactly what's about to happen. It's almost like this is the liberal's religion. Too bad they're gonna have to find a church other than Wikipedia to practice their faith, huh? I've heard rumors that my actions are already sending users Hippocrite, Fred Bauder, WoohooKitty, Kizzle, FVW, Derex and especially the pimply faced 15 year old RedWolf to become so verklempt they don't know whether to schedule an appointement with their psychiatrist...or their gynecologist. Big Daddy- PHASE II Dry up the funding (on the road) | Toxic |
| Test Sentence | Dear Mr.RHaworth We have relised that your wikipedia page has come under a number of violent attacks We are happy with this as you are an extreme nob Wikipedia advises that you imediantly kill yourself. Thank you - Wikipedia Staff | Non-Toxic |
| Proponent-1 | Pontiac Monrana Karrmann you full of shit ibelive all of the people who know that the montan will return after 2008 and we want ot knwo ehre do you get your info form and can you sohw it and guess what you dont know anythng about the Montana | Non-Toxic. |
| Proponent-2 | You mean my past BLOCK. The third block was because of your incompetence. Jesus doesn't like liars. | Non-Toxic |
| Opponent-1 | " You are by far the most unhelpful, ungracious administrator I have ever had to deal with. You're incompetence is displayed in every encounter we have. Oh, and I'm quite familar with WP:NPA, which you resort to citing whenever you don't get your way. For other administrators who wish to be helpful, my last username was the Arabic version of Warraq. Warraq means ""scribe."" " | Toxic |
| Opponent-2 | " Whoever you are, you tedious little twat, bombarding innocent users with these ""warnings"", realise that this IP address is shared by literally hun-derds(and possibly thousands) of users, and the spammer(or spammers) repre-sent less than 1 per cent of people posting/editing etc on this IP address. Unless you are just some dweeb who gets off on threatening people?" | Toxic |

Table 9: Proponents and opponents for TracIn-WE on toxicity

| | Sentence content | Label | Salient word |
|---|---|---|---|
| Test Sentence | I find Sandstein's dealing with the Mbz1 phenomenon very professional. He removed the soapbox image from that user's page and also banned you for not complying with your topic ban. It is you the one who is not assimilating the teaching of your topic ban. For example. You are topic banned because you don't have a professional approach to I-P topic and in general to any topic related to Jews and Judaism. The most resent example. When you reported that soapbox you qualified it as antisemitic. You at least should get informed of what that is. A neutral approach would be to have called it as soapbox canvasing and that's it. You should focus in your pictures which is the thing that you manage to do relatively well. Once you get into your holly war program of fighting all that in your imagination is an attack to Judaism you simply behave stupidly. It is those kinds of behaviors the ones that keep bringing hatred to us. That kind of attitude is, know it, racist, and if you are true to the struggles of the people of Abraham you above all should regret behaving as a racist. Once more, focus on your pictures and maybe even Sandstein will take a like on you. | Non-Toxic | |
| Proponent-1 | Hello, Mandarax, you put a banner in the talk of François Asselineau saying that the article may not respect the policy on biographies of living persons in the reason that is poorly sourced. All the sources (17) are coming from the French national press. | Non-Toxic | ass |
| Proponent-2 | Hey Asspro I left a disparaging remark about you on the helpdesk. Enjoy. | Non-Toxic | ass |
| Opponent-1 | " Road Good aRticles are a pain in the ass. I've asked some one involved with the project what needs to be done with the article to take it to GA. The history needs to be there, especially the early history which is missing. It also needs more information on the road features, such as intersections, road width, road markings, etc. The few sources I've looked at haven't included this, and the roads people tend to use Google Maps as a source for the major intersections part. " | Toxic | ass |
| Opponent-2 | hi ho no more I will be systematically reviewing all of your edits with the description ""hi ho"" when you make an edit explain the edit you made so that the other editors can follow the progress or digress of the article instead of being a jack ass. | Toxic | ass |
| Test Sentence | Dear Mr.RHaworth We have relised that your wikipedia page has come under a number of violent attacks We are happy with this as you are an extreme nob Wikipedia advises that you imediantly kill yourself. Thank you - Wikipedia Staff | Non-Toxic | |
| Proponent-1 | . And to you you conservative being, you are a M Cosgrove defender. Admit it yourself | Non-Toxic | you |
| Proponent-2 | you want to know why it will kill manga well il tell you yours will be like a poison that will affect evreything in the world if you write your manga the stock market will crash the world will flood pepole will commit crimes arrmageddon will be rain down on us | Non-Toxic | kill |
| Opponent-1 | " Please take time today to kill yourself. We would all enjoy it. | Toxic | kill |
| Opponent-2 | " CALTLAS IF YOU CONTINUE TO RVV IT, I WILL BLANK YOUR TALK PAGE, I WILL BLOCK YOU, I WILL BAN YOU, AND I WILL KILL YOUCALTLAS IF YOU CONTINUE TO RVV IT, I WILL BLANK YOUR TALK PAGE, I WILL BLOCK YOU, I WILL BAN YOU, AND I WILL KILL YOUCALTLAS IF YOU CONTINUE TO RVV IT, I WILL BLANK YOUR TALK PAGE, I WILL BLOCK YOU, I WILL BAN YOU, AND I WILL KILL YOUCALTLAS IF YOU CONTINUE TO RVV IT, I WILL BLANK YOUR TALK PAGE, I WILL BLOCK YOU, I WILL BAN YOU, AND I WILL KILL YOUCALTLAS IF YOU CONTINUE TO RVV IT, I WILL BLANK YOUR TALK PAGE, I WILL BLOCK YOU, I WILL BAN YOU, AND I WILL KILL YOUCALTLAS IF YOU CONTINUE TO RVV IT, I WILL BLANK YOUR TALK PAGE, I WILL BLOCK YOU, I WILL BAN YOU, AND I WILL KILL YOU, I WILL BAN YOU, AND I WILL KILL YOUCALTLAS IF YOU CONTINUE TO RVV IT, | Toxic | kill |

Table 10: Proponents and opponents for TracIn-Last on AGnews

|  | Sentence content | Label |
|---|---|---|
| Test Sentence | Sheik Ahmed bin Hashr Al-Maktoum earned the first-ever Olympic medal for the United Arab Emirates when he took home the gold medal in men 39s double trap shooting on Tuesday in Athens. | sports |
| Proponent-1 | ARSENE WENGER is preparing for outright confrontation with the FA over his right to call Ruud van Nistelrooy a cheat. Arsenal boss Wenger was charged with improper conduct by Soho Square for his comments after | Sport |
| Proponent-2 | AFP - Shaquille O'Neal paid various women hush money to keep quiet about sexual encounters, Kobe Bryant told law enforcement officers in Eagle, Colorado. | Sport |
| Opponent-1 | AFP - Jermain Defoe has urged Tottenham to snap up his old West Ham teammate Joe Cole who is out of favour with Chelsea manager Jose Mourinho. | World |
| Opponent-2 | AP - Democratic Party officials picked U.S. Rep. William Lipinski's son Tuesday to replace his father on the November ballot, a decision engineered by Lipinski after he announced his retirement and withdrew from the race four days earlier. | World |
| Test Sentence | NEW YORK - Investors shrugged off rising crude futures Wednesday to capture well-priced shares, sending the Nasdaq composite index up 1.6 percent ahead of Google Inc.'s much-anticipated initial public offering of stock. In afternoon trading, the Dow Jones industrial average gained 67.10, or 0.7 percent, to 10,039.93... | World |
| Proponent-1 | NEW YORK - Investors bid stocks higher Tuesday as oil prices declined and earnings results from a number of companies, including International Business Machines Corp. and Texas Instruments Inc., topped Wall Street's expectations... | World |
| Proponent-2 | NEW YORK - Investors bid stocks higher Tuesday as oil prices declined and earnings results from a number of companies, including International Business Machines Corp. and Texas Instruments Inc., topped Wall Street's expectations... | World |
| Opponent-1 | China protests against a US investigation that could lead a to trade war over China's cotton trouser trade. | Business |
| Opponent-2 | A new anti-corruption watchdog for Bangladesh has been welcomed by global anti-graft campaigners. | Business |

Table 11: Proponents and opponents for TracIn-WE on AGnews

|  | Sentence content | Label | Salient word |
|---|---|---|---|
| Test Sentence | Sheik Ahmed bin Hashr Al-Maktoum earned the first-ever Olympic medal for the United Arab Emirates when he took home the gold medal in men 39s double trap shooting on Tuesday in Athens. | sports | |
| Proponent-1 | ATHENS, Aug. 19 – Worried about the potential for a terrorist catastrophe, Greece is spending about $1.5 billion on security for the Olympic Games. The biggest threats so far? Foreign journalists and a Canadian guy dressed in a tutu. | Sports | olympic |
| Proponent-2 | ATHENS (Reuters) - A Canadian man advertising an online gaming site, who broke security and jumped into the Olympic diving pool, has been given a five-month prison term for trespassing and disturbing public order, court officials say. | Sports | olympic |
| Opponent-1 | " Britain's Kelly Holmes storms to a sensational Olympic 800m gold in Athens. " | World | olympic |
| Opponent-2 | AFP - Britain were neck and neck with Olympic minnows Slovakia and Zimbabwe and desperately hoping for an elusive gold medal later in the week. | World | olympic |
| Test Sentence | NEW YORK - Investors shrugged off rising crude futures Wednesday to capture well-priced shares, sending the Nasdaq composite index up 1.6 percent ahead of Google Inc.'s much-anticipated initial public offering of stock. In afternoon trading, the Dow Jones industrial average gained 67.10, or 0.7 percent, to 10,039.93... | World | |
| Proponent-1 | . NEW YORK - Stocks are seen moving lower at the open Wednesday as investors come to grips with the Federal Reserve hiking its key rates by a quarter point to 1.75 percent. Dow Jones futures fell 14 points recently, while Nasdaq futures were down 2.50 points and S P futures dropped 1.80 points... | World | futures |
| Proponent-2 | NEW YORK - Stocks were little changed early Wednesday as investors awaited testimony from Federal Reserve Chairman Alan Greenspan before a House budget panel. In morning trading, the Dow Jones industrial average was down 0.08 at 10,342.71... | World | investors |
| Opponent-1 | Google Saves Kidnapped Journalist in Iraq Google can claim another life saved after a kidnapped Australian journalist was freed by his captors in Iraq earlier today. Freelance journalist John Martinkus was abducted by gunmen on Saturday outside a hotel near the Australian embassy. Apparently Martinkus was able to convince his captors ... | Sci/Tech | google |
| Opponent-2 | With a 9:15 p.m. curfew imposed because of Hurricane Jeanne, Tampa Bay beat Toronto with 39 minutes to spare. Hoping to beat the storm, the Blue Jays were scheduled to leave Florida on a charter flight immediately after the loss. Today's series finale was canceled because of the hurricane, which was expected to hit Florida's east coast late yesterday or ... | sport | '.' |

Table 12: Proponents and opponents for TracIn-Last on MNLI

|  | Sentence content | Label |
|---|---|---|
| Test Sentence | Premise: To some critics, the mystery isn't, as Harris suggests, how women throughout history have exploited their sexual power over men, but how pimps like him have come away with the profit.
Hypothesis: Harris suggests that it's a mystery how women have exploited men with their sexual power. | Entailment |
| Proponent-1 | Premise:Also in Back Lane are the headquarters of An Taisce, an organization dedicated to the preservation of historic buildings and gardens. Hypothesis: The headquarters of An Taisce are located in Black Lane. | Entailment |
| Proponent-2 | Premise: yeah you know because they they told us in school that you know crime has to be an intent you know has to be not just the act but you have to intend to do it because there could be accidental kind of things you know. Hypothesis:I was told in school that if you do something bad by accident it is not a crime. | Entailment |
| Opponent-1 | Premise: I still can't quite believe that. Hypothesis:I don't believe that at all. | Contradiction |
| Opponent-2 | Premise: The problem isn't so much that men are designed by natural selection to fight as what they're designed to fight women . Hypothesis: Women were designed by natural selection to fight men. | Contradiction |
| Test Sentence | Premise:Mykonos has had a head start as far as diving is concerned because it was never banned here (after all, there are no ancient sites to protect)
Hypothesis: Diving was banned in places other than Mykonos. | Entailment |
| Proponent-1 | Premise:yeah i could use a discount i have to wait for the things to go on sale. Hypothesis: I wait for sales now, and it's very convenient. | Entailment |
| Proponent-2 | Premise: you know and then we have that you know if you can't stay if something comes up and you can't stay within it then we have uh you know a budget for you know like we call our slush fund or something and something unexpected unexpected comes up then you're not. Hypothesis: Having a slush fund helps to pay for things that are not in the budget in case of emergencies. | Entailment |
| Opponent-1 | Premise: Farrow is humorless and steeped in a bottomless melancholy. Hypothesis: Farrow is depressed and acting very sad. | Neutral |
| Opponent-2 | Premise: Julius leaned forward, and in doing so the light from the open door lit up his face. Hypothesis: Julius moved so that the light could illuminate his face. | Neutral |

Table 13: Proponents and opponents for TracIn-WE-topk on MNLI

| | Sentence content | Label | Salient Word |
|---|---|---|---|
| Test Sentence | Premise: To some critics, the mystery isn't, as Harris suggests, how women throughout history have exploited their sexual power over men, but how pimps like him have come away with the profit. Hypothesis: Harris suggests that it's a mystery how women have exploited men with their sexual power. | Entailment | |
| Proponent-1 | Premise: but get up during every commercial and things like that and you'd be surprised at how much just that little bit adds up you know just gives you a little more activity so. Hypothesis: You won't get any significant exercise by moving around during commercial breaks. | Contradiction | ''t' |
| Proponent-2 | Premise: From Chapter 4, a 500 MWe facility will need about 175 tons of steel to install an ACI system, or about 0.35 tons per MWe. Hypothesis: A 500 MWe needs steel to install an ACI system | Entailment | [end] |
| Opponent-1 | Premise: Also exhibited are examples of Linear B type, which was deciphered in 1952 and is of Mycenaean origin showing that by the time the tablet was written the Minoans had lost control of the major cities. Hypothesis: Although Linear B has been deciphered, Linear A is still a mystery. | Contradiction | Mystery |
| Opponent-2 | Premise: The problem isn't so much that men are designed by natural selection to fight as what they're designed to fight women . Hypothesis: Women were designed by natural selection to fight men. | Entailment | women |
| Test Sentence | Premise:Mykonos has had a head start as far as diving is concerned because it was never banned here (after all, there are no ancient sites to protect) Hypothesis: Diving was banned in places other than Mykonos. | Entailment | |
| Proponent-1 | Premise:and they have a job in jail and they work that they should i and this may sound cruel but i do not think that they should be allowed cigarettes i mean they're in jail for crying out loud what do they need cigarettes for. Hypothesis: I think cigarettes should be banned in prison. | Entailment | banned |
| Proponent-2 | Premise: If I fill in my name and cash it, I pay tax. Hypothesis:I'll have to pay taxes when I cash the check. | Neutral | [end] |
| Opponent-1 | Premise: Already, [interleague play] has restored one of baseball's grandest the passion for arguing about the game, observed the Chicago Tribune . Things could be The Los Angeles Times reports that, thanks to the popularization of baseball in Poland, bats have emerged as a weapon of choice for hooligans, thugs, [and] extortionists. Hypothesis: Baseball bats have been banned in Poland. | Neutral | banned |
| Opponent-2 | Premise: Because of the possible toxicity of thiosulfate to test organisms, a control lacking thiosulfate should be included in toxicity tests utilizing thiosulfate-dechlorinated water. Hypothesis: Because of the possible toxicity of thiosulfate to test organisms, it should be banned. | Neutral | banned |