# OpenReview forum: "First is Better Than Last for Language Data Influence"
_NeurIPS.cc/2022/Conference — NeurIPS 2022 Accept_

### Official Review · Reviewer_R7Dm · 2022-07-03

**Rating:** 8
**Confidence:** 4
**Soundness:** 3 good
**Presentation:** 4 excellent
**Contribution:** 4 excellent

**Summary:**

This paper thoroughly examines the effectiveness of different parameters serving as data influence measurement. It provides two important insights on the informativeness of parameters in lower layers and cancellation effect of parameters used by more training examples. Motivated by these insights, it develops a novel data influence method by replacing parameters of the last layer used in TracIn with embedding layer. Empirical results verify the insights and the effectiveness of the proposed method.

**Questions:**

* Is it essential to use *gradient-based* similarity as the similarity term $S$?
* Some typos:
  * line 36: parametersin -> parameters in
  * line 194: "share" logic -> "share logic"
  * line 215: multiple periods.
  * Appendix Figure 4: or -> on


**Limitations:**

* The proposed method only works well when word embedding is fixed.

**Strengths And Weaknesses:**

**Strengths**
* This paper provides two valuable insights:
  * First, the gradients of the embedding weights capture high-level information due to gradient chain.
  * Second, parameters used by more training examples suffer more from 'cancellation effect' (i.e. gradients of multiple instances have large magnitude, but different signs).
* The proposed method takes both low- and high-level similarity into consideration.
* Good writing and clear presentation.

**Weakness**
* The terms of 'share logic' and 'unique logic' are vague. Could you provide explanations and evidence?
* There should be an ablation study for using special tokens only on full dataset to demonstrate the effectiveness of word overlap.
* Removing data points also change the training process (i.e. schedule of batches). How do you control related variables to be consistent for different methods? Does the random seed of data sampler influence the model performance significantly?

---

> ### Author Response · Authors · 2022-08-01
> **Response to official review by reviewer R7Dm**
>
> We thank the reviewer for the constructive reviews.
>
> – Unique logic VS shared logic:
>
> One way to quantify the shared logic level of weight w is to measure the expected gradient similarity, E_xa, xb {COS_SIM [dl(xa)/ w , dl(xb)/ w]}, where COS_SIM is the cosine similarity. This measures the expected gradient cosine similarity between two examples. The expected gradient similarity for testing examples between different layers in the CNN classification are: first 0.035, second 0.075, third 0.21, last 0.23. This verifies that the latter layers in the neural network have more aligned gradients between examples, and thus share more logics between training examples.
>
> – ablation study for using special tokens only:
>
> The AUC-DEL+ and AUC-DEL- for special tokens (CLS and SEP) only on AGnews is -0.029, 0.018, and the AUC-DEL+ and AUC-DEL- for special tokens (CLS and SEP) only on Toxic is 0.016 and 0.014. The special tokens alone perform much worse than TracIn-WE as they do not consider the information for overlapping words, validating the importance of overlapping words.
>
> – How do you control related variables to be consistent for different methods? Does the random seed of data sampler influence the model performance significantly?
>
> For fair comparisons, we always do a random shuffle before the training of each batch. We also average the retraining result over 10-runs to reduce the randomness of the random seed (while we do not manually set the random seed in our experiments).
>
> – Is it essential to use gradient-based similarity as the similarity term:
>
> We have tested using TF-idf similarity as the similarity term, but while it performs well on simpler tasks such as AGnews and Toxic, it performs very badly on mNLI (worse than TracIn-Last). Our interpretation of this result is that similarity terms based on gradients can better capture task-related semantic information , while similarity terms based on word similarity may be only useful when the task is highly-related to word similarity (AGnews and Toxic are very dependent on word similarity, but not mNLI). There might be other model-dependent similarity terms that make sense, but currently we do not know any other good similarity terms to get good influence scores.
>
> – The proposed method only works well when word embedding is fixed:
>
> Just to clarify, when we only want to find influential training examples without word overlap, both the proposed TracIn-WE and any other baselines (influence function, TracIn, Representer point) do not work well (see Fig. 4 in appendix) when word embedding is not fixed. On the contrary, when we want to find any influential training examples, the proposed TracIn-WE works very well under both fixing and training the embeddings settings. This hints that when the word embedding is trainable, sentences with no word overlaps do not carry strong influential signals. This is not a limitation of TracIn-WE but instead a phenomenon for all training data influence methods that we know of.

---

> ### Author Response · Authors · 2022-08-06
> **Thanks for the review**
>
> Dear reviewer,
>
> We would like to thank you for your feedback. We hope that our rebuttal has addressed any concerns you may have for our paper. If you have any unresolved concerns, please let us know so we could try to address them during the author-reviewer discussion period (ending following Tuesday).

---

### Official Review · Reviewer_iZ6x · 2022-07-11

**Rating:** 5
**Confidence:** 4
**Soundness:** 2 fair
**Presentation:** 2 fair
**Contribution:** 2 fair

**Summary:**

The paper proposes TracIn-WE that will produce data influence scores based on the level of the overall training input and at the level of words within the training input. The method mitigates an effect called “cancellation effect” resulting from the data influence extraction methods based the last layer weights. The word-embedding layer based proposed method (TracIn-WE) is built on the existing method called TracIn which is constructed on the last layer.

**Questions:**

What is the interpretation of cancellation ratio?



**Limitations:**

They have discussed limitations.


**Strengths And Weaknesses:**

Strengths:

The method is based on the observation of existing data influence method’s cancellation across the data influence attributions (i.e., distributed test data loss) to training examples, i.e., the sign of attributions across training examples disagree and cancels each other out.  This kind of observation might be useful to develop improved methodologies for explainable NLP.

Weakness

1.	Example 3.1 is not clear to me as the example is based on a sparse representation whereas the word embedding inputs are dense representations.

2.	The word embeddings provide the input representation. As a result, the similarity of inputs (train and test) would better capture by the embedding space. However, the data’s influence on a prediction is related to the output layers (last layer) which is closely linked to loss function. Using embedding space for data influence extraction has some limitations in terms of deep neural network training (e.g., vanishing gradient problem). It is not clear how the proposed method solves those fundamental problem (e.g., learning and loss relevance) when calculating the data influence based on the embedding layer.

3.	Bias plays different rules than other weight. Removing Bias from the TracIn Calculation to Reduce Cancellation Effect needs further justification. What is the interpretation of cancellation ratio?

---

> ### Author Response · Authors · 2022-08-01
> **Response to official review by reviewer iZ6x**
>
> We thank the reviewer for the constructive reviews.
>
> – example 3.1 – sparse representation VS dense representations:
>
> The key point of Example 3.1 is that gradient updates from individual examples to the weight vector w are sparse, and are less likely to cancel each other out. On the other hand, gradient updates to the bias are likely to cancel each other out as all training examples update the bias. Similarly, updates to word (token) embedding parameters are also sparse in that each input sentence only updates the parameters of tokens present in the sentence. Concretely, if the vocabulary size is 25000 and the embedding dimension is 128 then the embedding matrix contains 25000 * 128 trainable parameters. For each input sentence with length 128, at most 128*128 weight parameters will be updated by the resulting weight gradients. Thus, each embedding parameter is only updated by 128/25000 of the training data on average. This is quite sparse (in terms of the vocabulary dimension 25000) in comparison to the bias parameters which are updated by all training data. Thus, the updates to the bias parameter would be subject to a stronger cancellation effect than updates to the embedding parameters.
>
> – potential vanishing gradient problem using gradient of word embeddings:
>
> We do not observe gradient vanishing issues for word embedding gradients, mainly because modern deep neural networks (self-attention based architectures) contain residual blocks and ReLU activations to prevent gradient vanishing. If the model suffers from gradient vanishing, using later layers for data influence should have better deletion curve compared using the first layer as the gradient from the first layer may have reductive information; however, we observe that the first layer performs better than later layers for both self-attention models and CNN architectures, hinting that the gradient to word embeddings does not suffer from vanishing gradient.
>
> – It is not clear how the proposed method solves those fundamental problems (e.g., learning and loss relevance) when calculating the data influence based on the embedding layer.
>
> The loss gradient of the word embedding layers contains the information of loss by chain law. As dL(x_t)/dw = dL(x_t)/df(x_t) * df(x_t)/dw, the loss gradient with respect to w contains a term that is solely based on the loss, and thus is relevant to the loss.
>
> – What is the interpretation of cancellation ratio? Why remove Bias for influence calculation?
>
> The cancellation ratio is the ratio of A=|(sum of magnitude of influence caused by the weight parameters across training examples and testing examples)| and B=|(sum of influence caused by the weight parameters across training examples and testing examples)|.
>
> Interpretation of A:
> The absolute total influence amount change caused by the weight parameters. If A is large, the weight parameter greatly affects the influence scores.
>
> Interpretation of B:
> The absolute total estimated loss change of the testing examples caused the weight parameters after training. If B is small, this means that the weight parameters are not crucial to the change of loss.
>
> When the cancellation ratio is large (such that A >>> B), this means that a non-crucial weight parameter W greatly changes the influence score, which may not be ideal. Applying this interpretation to the cancellation of bias parameters, the intuition is that the bias parameters are not mainly responsible for the reduction of testing example loss change during the training process (since term B is small). However, they dominate the total influence strength due to their dense nature (term A is large). We could consider the following motivating example:
>
> Assume in the training process, the batch size is 2, and one positive example and one negative example are always guaranteed for each mini-batch of data. Assume that during the training, the bias parameter is always 0 since the positive example and negative example gradient for the bias will always cancel each other out. In this scenario, the bias should not have any impact on the loss change, and term B will be 0. However, term A will still be large, as the influence of the positive example and the influence of the negative example are both non-zero, but they cancel each other out so that term B is 0. This is an example where even though the bias does not affect the model at all, it affects the actual influence distribution, which leads to infinite cancellation score. The influence score change caused by the bias would not be meaningful, and should be removed.

---

> ### Author Response · Authors · 2022-08-06
> **Thanks for the review**
>
> Dear reviewer,
>
> We would like to thank you for your feedback. We hope that our rebuttal has addressed any concerns you may have for our paper. If you have any unresolved concerns, please let us know so we could try to address them during the author-reviewer discussion period (ending following Tuesday).

---

> ### Author Response · Authors · 2022-08-09
> **Response discussion**
>
> Dear reviewer,
>
> We hope that our response clarifies our motivation example 3.1 and the meaning of cancellation. If there are any remaining questions we hope to have a chance to answer before discussion deadline.

---

### Official Review · Reviewer_LSEU · 2022-07-11

**Rating:** 6
**Confidence:** 3
**Soundness:** 3 good
**Presentation:** 3 good
**Contribution:** 3 good

**Summary:**

This paper revisits the common practice of computing training data influence, especially methods for computing the influence only from the last layer and investigates improving the estimation of examples' influence.
The model size used in the NLP field is huge because we often incorporate a pre-trained language model as a base model.
We usually compute influence functions only from the final layers.
This paper also reveals that last layer representations in text classification models may suffer from the cancellation effect when computing the influence function on training samples.
This paper proposes a technique called TracIn-WE to mitigate the cancellation effect.
TracIn-WE is a modification of the existing method, TracIn, to compute the influence of each sample on the word embedding layer instead of the last layer.
This paper conducts experiments on three text classification tasks and shows that TracIn-WE significantly outperforms other data influence methods applied on the last layer.


**Questions:**

* Have you ever tried to visualize the reducing cancellation effect for each layer?

**Limitations:**

This paper has limitation and potential negative societal impact sections in Appendix.
There are no additional concerns.

**Strengths And Weaknesses:**

Strengths:
* This paper reveals the issues of existing influence computation methods by calculated from the last layer.
* The empirical evaluation shows the better performance on the proposed method.
* This paper tackles the important research theme involved in XAI.


Weaknesses:
* The method is actually a straight forward extension, so that the technical novelty is marginal.

---

> ### Author Response · Authors · 2022-08-01
> **Response to official review by reviewer LSEU**
>
> We thank the reviewer for the constructive reviews.
>
> – A straightforward extension lacks novelty:
>
> ​​The main novelty/contribution of our paper lies in making two observations: (1) Using the last layer to calculate data influence suffers from cancellation and leads to an inferior measure of influence (as evidenced by our case deletion eval), and (2) Using the first layer mitigates the issue, and leads to a significantly better measure of influence (in some cases >10 times on the "case deletion" evaluation).  (1) is surprising as restricting to the last layer is the most popular choice for approximating influence computation and widely used, yet this issue has not be identified before. While the contribution (2) seems to be a straightforward extension, it is actually surprising as one does not expect the first (embedding) layer to carry high-level / semantic information, and thus such a simple extension has never been considered in the field. Fortunately, first layer gradients chain sufficient semantic information from higher layer layers to offer a meaningful measure of influence; see Table 2 for examples. We argue that the simplicity of our extension is actually a strength of the paper as it addresses a significant existing issue that was not identified before.
>
>
> —Have you ever tried to visualize the reducing cancellation effect for each layer?
>
> We added a cancellation by layers plot in Fig.5 of appendix.

---

> ### Author Response · Authors · 2022-08-06
> **Thanks for the review**
>
> Dear reviewer,
>
> We would like to thank you for your feedback. We hope that our rebuttal has addressed any concerns you may have for our paper. If you have any unresolved concerns, please let us know so we could try to address them during the author-reviewer discussion period (ending following Tuesday).

---

### Meta-Review · Area_Chair_e7HG · 2022-08-28

**Recommendation:** Accept
**Confidence:** Certain

**Metareview:**

This paper studies an important problem, of identifying influential training examples. It exposes a potential shortcoming in prior work, of focusing on the last layer, and proposes an alternative method. The approach cleverly assures looking at word overlap via overlapping word embeddings while still aggregating high level information from the back flowing gradients. The reviewers appreciated the topic, the insights and observations, and the empirical observations.  Overall this paper is making novel contributions to an important area.

There was some worry of novelty but I agree that the findings are novel and meaningful. The reviewers also raised various questions about vagueness of terms, which the authors addressed in the their response. There were also comments on missing controls and ablations, which the authors partly addressed in their response. To this, during the discussion, a suggestion was made to "to add the variance of multiple runs or a significant test, since the robustness towards randomness is also very important to a reliable data influence measurement". I strongly agree.

Some technical questions by Reviewer iZ6x were answered. Please make sure to include the answers to them in the next revision, as well as all the clarifications and ablations provided in the author responses.

Finally, I would strongly suggest adding experiments with at least one stronger model besides BERT, such as RoBERTa or DeBERTa. This would help give confidence that the approach is relevant for newer models.

AC

**Award:**

No

---

### Decision · Program_Chairs · 2022-09-14

Accept